# Maize PPR278 Functions in Mitochondrial RNA Splicing and Editing

**DOI:** 10.3390/ijms23063035

**Published:** 2022-03-11

**Authors:** Jing Yang, Yang Cui, Xiangbo Zhang, Zhijia Yang, Jinsheng Lai, Weibin Song, Jingang Liang, Xinhai Li

**Affiliations:** 1National Engineering Laboratory for Crop Molecular Breeding, Institute of Crop Science, Chinese Academy of Agricultural Sciences, Beijing 100081, China; yangjing@cau.edu.cn; 2State Key Laboratory of Plant Physiology and Biochemistry, China Agricultural University, Beijing 100193, China; september516@163.com (Y.C.); zhxiangbo@126.com (X.Z.); jiageyang@163.com (Z.Y.); jlai@cau.edu.cn (J.L.); songwb@cau.edu.cn (W.S.); 3National Maize Improvement Center, Department of Plant Genetics and Breeding, China Agricultural University, Beijing 100193, China; 4Development Center of Science and Technology, Ministry of Agriculture and Rural Affairs, Beijing 100176, China

**Keywords:** PPR protein, intron splicing, C-to-U RNA editing, maize, seed development

## Abstract

Pentatricopeptide repeat (PPR) proteins are a large protein family in higher plants and play important roles during seed development. Most reported PPR proteins function in mitochondria. However, some PPR proteins localize to more than one organelle; functional characterization of these proteins remains limited in maize (*Zea mays* L.). Here, we cloned and analyzed the function of a P-subfamily PPR protein, PPR278. Loss-function of *PPR278* led to a lower germination rate and other defects at the seedling stage, as well as smaller kernels compared to the wild type. *PPR278* was expressed in all investigated tissues. Furthermore, we determined that PPR278 is involved in the splicing of two mitochondrial transcripts (*nad2* intron 4 and *nad5* introns 1 and 4), as well as RNA editing of C-to-U sites in 10 mitochondrial transcripts. PPR278 localized to the nucleus, implying that it may function as a transcriptional regulator during seed development. Our data indicate that PPR278 is involved in maize seed development via intron splicing and RNA editing in mitochondria and has potential regulatory roles in the nucleus.

## 1. Introduction

Pentatricopeptide repeat (PPR) proteins are widely distributed in plants, with more than 400 members in maize (*Zea mays* L.) [1], which were also divided into two subfamilies: The P subfamily containing motif P only, and the PLS subfamily harboring the P, L, and S motifs [2,3]. PLS-subfamily members were further classified into PLS, E, E+, and DYW types according to their C-terminal domain [4,5]. PPR proteins have been isolated from species including maize, rice (*Oryza sativa* L.), and *Arabidopsis thaliana* L., and functional analysis revealed that they could bind RNA and might play important roles in RNA metabolism, including RNA splicing and editing, cleavage, and maturation [3].

Ongoing work has shown that PPR proteins splice 22 group II introns present within 8 protein-coding genes in mitochondria in maize, including NADH dehydrogenase subunits1, 2, 4, 5, and 7 (*nad1*, *2*, *4*, *5*, *7*), and subunits of a complex involved in the biogenesis of cytochrome c: subunits F (*ccmF_C_*), cytochrome c oxidase subunits2 (*cox2*), and ribosomal protein s3 (*rps3*) [6,7,8]. Among these spliced genes, *nad1* introns 1, 3, and 4; *nad2* intron 2; and *nad5* introns 2 and 3 are post-transcriptionally modified via trans-splicing, while other introns of target spliced transcripts are modified through cis-splicing [6]. Loss-function of PPR proteins and/or other spliceosomes results in defects in seed and/or seedling development, such as an empty pericarp in the seed, decreased 100-kernel weight, and seedling death, due to abnormal intron splicing of mitochondrial genes, which leads to abnormal assembly of mitochondrial complex Ⅰ and insufficient production of energy molecules [9,10,11,12,13,14,15,16,17,18,19,20,21,22,23,24,25,26]. These findings highlight the importance of PPR-based RNA splicing in mitochondria for seed development in maize. 

Cytidine-to-uridine (C-to-U) RNA editing in mitochondria is an important function of PPR proteins and is required for correcting mutations at the RNA level by recognizing specific transcripts through tandem repeat motifs [27,28]. The DYW domain at the C terminus of some PPR proteins confers catalytic activity during editing; PPR proteins that lack this domain may recruit co-factors and form a larger complex to perform editing functions [4,29]. Indeed, studies in Arabidopsis, rice, and maize have confirmed that PLS-type PPR proteins function as cytidine editors in transcripts of mitochondrial and plastid genes [30]. For example, mutations in *Empty pericarp 5* (*EMP5*), *EMP7*, *EMP9*, *EMP18*, *EMP21*, *Defective kernel 10* (*DEK10*), *DEK36*, *DEK39*, *PPR2263*, *Small kernel 1* (*SMK1*), *SMK4*, and *SMK6* significantly altered the C-to-U editing efficiency in mitochondrial RNAs, leading to small or defective kernels [12,13,31,32,33,34,35,36,37,38,39,40,41].

More than 99% of PPR proteins across the *Arabidopsis* genome are predicted to localize in mitochondria and/or plastids, and very few localize in both nuclei and mitochondria [4,5,42]. SUPPRESSOR OF THE ABSCISIC ACID RECEPTOR OVEREXPRESSOR1 (SOAR1, a native regulator of abscisic acid signaling) and GLUTAMINE-RICH PROTEIN23 (GRP23) localize in both nuclei and mitochondria. GRP23 interacts with RNA polymerase II subunit III and PPR PROTEIN LOCALIZED TO NUCLEI AND MITOCHONDRIA1 (PNM1), as well as with the plant-specific Teosinte Branched 1/Cycloidea/Proliferating Cell Factor (TCP) family transcription factors TCP8 and nucleosome assembly protein (NAP1) in nuclei, indicating that this PPR protein likely functions as a regulator in nuclei [43,44,45]. In addition, rice NUCLEAR-LOCALIZED PPR PROTEIN1 (OsNPPR1) and OsNPPR3 also target to the nucleus, where they affect pre-mRNA alternative splicing of nuclear genes related to mitochondrial function [46,47]. Furthermore, DEK43 targets to mitochondria and nuclei, but its functions in the nuclei remain to be investigated [19]. Thus, further study is needed to uncover the functions of PPR proteins with multiple-organelle localization.

In this study, we isolated PPR278, a P-subfamily PPR protein with dual localization in nuclei and cytoplasm of maize. We discovered that splicing of *nad2* and *nad5* introns and C-to-U editing of 10 mitochondrial transcripts were affected in the *ppr278* mutants. RNA-sequencing (RNA-seq) analysis revealed that the expression of nucleus-encoded genes associated with RNA splicing and editing was significantly altered in the *ppr278* mutants. Our findings demonstrate that PPR278 affects seed development via effects on RNA intron splicing and editing in mitochondria and on gene expression in nuclei, expanding our limited knowledge about the function of P-subfamily PPR proteins that are capable of splicing introns.

## 2. Results

### 2.1. The ppr278-1 Mutants Cause Aberrant Seed Development in Maize

We isolated a kernel mutant from our mutant libraries established by an ethyl methanesulfonate (EMS)-based method, which produced smaller seeds compared to the wild type (WT) (Figure 1A–D), and named it *ppr278-1* based on similar phenotypes with other *ppr* mutants in maize. Only ~5% of mutant kernels could germinate normally on average and grew to the seedling stage (Figure 1E and Appendix A). However, the vigor of mutant seedlings was notably decreased, and homozygous mutant seedlings did not survive to the reproductive stage (Appendix A). The 100-kernel weight of *ppr278* mutant plants was significantly lower than that of WT, which might be due to developmental defects in mutant kernels or disruption of seed-filling processes (Figure 1F,G). The segregation ratio of seeds in mutant ears was 3:1 (normal kernels: mutant kernels, 212:69), implying that the *ppr278-1* phenotype is controlled by a single recessive locus (Figure 1A,B). 

### 2.2. PPR278 Encodes a Constitutively Expressed PPR Protein

We used bulked segregate analysis-RNA sequencing (BSR) to isolate PPR278. Endosperm and embryos from M3 generation kernels at 12 days after pollination (DAP) were used to construct RNA-seq libraries of WT and mutant-type pools. High-quality single nucleotide polymorphism (SNP) markers were called by aligning RNA-seq reads to the B73 reference genome (V4). We identified 32 candidate genes including 14 missense variants, 10 synonymous variants, 5 3′ untranslated region (UTR) variants, 2 5′UTR variants, and 1 stop codon-gained variant (*Zm00001d015156*) (Appendix A). *Zm00001d015156* (located on chromosome 5) that had the highest ΔSNP index among missense variants and the stop codon-gained variant [48]; therefore, we determined *Zm00001d015156* as the candidate gene corresponding to PPR278. Sequence analysis revealed that *Zm00001d015156* had one SNP (C-to-T mutation) in the *ppr278-1* mutant. This SNP was located in the 15th exon at +2269 bp from the ATG start codon, causing a premature stop codon in *Zm00001d015156* (Figure 2A). To test whether the mutation in *Zm00001d015156* causes the *ppr278-1* phenotypes, we performed co-segregation analysis by genotyping a larger segregating population (281 kernels) and observed that the SNP in the *ppr278-1* mutant co-segregated with the low seed germination phenotype (Appendix A). To further test the candidate gene, we screened another mutant (hereafter referred to as *ppr278-2*) from the mutant libraries, which showed a similar phenotype as the *ppr278-1* mutant (Appendix A). We identified a SNP (C-to-T mutation) at +2404 bp in the 15th exon of *Zm00001d015156* in the *ppr278-2* mutant, also resulting in a premature stop codon (Figure 2A). Crossing, self-pollinating, and subsequent genetic analyses confirmed *ppr278-1* and *ppr278-2* as allelic mutants, revealing that *Zm00001d015156* is the causal gene for PPR278 (Figure 2B). Expression analysis showed that PPR278 was expressed in multiple tissues, with the highest expression level in 6-DAP kernels (Figure 2D). Thus, we cloned PPR278, encoding a constitutively expressed PPR protein, and confirmed that loss-of-function alleles of PPR278 caused the abnormal phenotype mentioned above.

### 2.3. PPR278 Belongs to the p-Subfamily and Dually Localizes in Cytoplasm and Nucleus

PPR protein sequences were downloaded from either NCBI (https://www.ncbi.nlm.nih.gov/) (Accessed on 9 March 2022) or PPR energy (https://ppr.plantenergy.uwa.edu.au/) (Accessed on 9 March 2022) databases, and full-length protein sequences were used for phylogenetic analysis, in which PPR278 was clustered into the P-subfamily (Appendix A) [49,50]. The prediction of PPR energy showed that *PPR278* and its *Arabidopsis* homolog *AT4G21880.1* grouped into the P-subfamily clade [42]. Further analysis indicated that PPR278 contained 11 *p* motifs (Figure 2C, Appendix A), and amino acid numbers of these motifs ranged from 32 to 37, which differed from typical P-subfamily PPR proteins (Appendix A).

Most PPR proteins target to mitochondria and/or plastids in plants, whereas only a few are predicted to localize in nuclei [4,5,42]. Based on Csbio (http://www.csbio.sjtu.edu.cn/bioinf/Cell-PLoc-2/) (Accessed on 9 March 2022) and Cello (http://cello.life.nctu.edu.tw/) (Accessed on 9 March 2022), PPR278 was predicted to be in the nucleus and cytoplasm (Appendix A). Subcellular localization experiments using onion (*Allium cepa* L.) epidermal cells and maize protoplasts revealed that PPR278 targeted to both nuclei and cytoplasm, suggesting that the functions of PPR278 are more complex than those of other PPR proteins targeted to mitochondria or/and plastids (Figure 3A,B). Moreover, the secondary structure of PPR278 consists of numerous paired anti-parallel alpha helices, as predicted by the AlphaFold Protein Structure Database (http://www.alphafold.ebi.ac.uk) (Accessed on 9 March 2022) (Figure 3C), which form the helix-turn-helix motifs necessary for RNA binding [5,51]. Therefore, even though PPR278 is a P-subfamily member, it contains 11 non-canonical P-type PPR repeats, which is a special feature. Another feature of PPR278 is that it is double-targeted in mitochondria and the nucleus. Like other PPR proteins, a series of paired anti-parallel alpha helices of PPR278 provide the ability to bind specific target RNA.

### 2.4. PPR278 Is Essential for the Cis-Splicing of nad2 Intron 4 and nad5 Introns 1 and 4

PPR proteins are RNA-binding proteins involved in RNA metabolism, including intron splicing and RNA editing, in mitochondria and plastids [3]. To test whether PPR278 is involved in cis-splicing in mitochondria, we performed semi-quantitative reverse transcription PCR (RT-PCR) analysis between mutant and WT plants for 35 mitochondrial genes using 12-DAP kernels. We observed expression changes in four genes, *rps2A*, *nad5*, *ccmF_N_*, and *apocytochrome b* (*cob*), between *ppr278-1* and WT plants (Figure 4A). Additionally, RT-PCR analysis showed that the cis-splicing efficiency of *nad2* intron 4 (1383 bp) and *nad5* introns 1 (870 bp) and 4 (952 bp) was affected in the *ppr278-1* and *ppr278-2* mutants (Figure 4B,C and Appendix A). We detected no differences at the transcript level or in cis-splicing for the other mitochondrial genes examined, including 25 tRNAs and 3 rRNAs between *ppr278* and WT plants (Appendix A). PPR278 is not only involved in mitochondrial gene expression (*rps2A*, *nad5*, *ccmF_N_*, and *cob*), but it is also necessary for the cis-splicing of *nad2* intron 4 and *nad5* introns 1 and 4.

### 2.5. PPR278 Is Required for C-to-U RNA Editing in 10 Mitochondrial Transcripts 

RNA editing is a major function of PPR proteins in mitochondria, and nearly 500 C-to-U RNA editing sites in mitochondrion-encoded transcripts were reported in the maize inbred line B73 [35]. Therefore, we examined the editing profiles of the 35 mitochondrion-encoded transcripts in the *ppr278* mutants and identified 94 C-to-U RNA editing events (Table 1, Appendix A). We analyzed the C-to-U RNA editing efficiency by directly comparing editing profiles in WT and *ppr278* plants and observed decreased efficiency of 67 C-to-U RNA editing sites in seven mitochondrial transcripts and increased editing efficiency in one editing site in *rps2B*-550 in the *ppr278* mutant (Table 1, Appendix A). However, we could not determine the editing efficiency of the remaining 26 C-to-U editing sites in *atp8* and *nad3* via the peak of C and/or U (Figure 5A and Appendix A). To understand how *atp8* and *nad3* are affected in the *ppr278* mutant, we sequenced ~40 independent clones from WT and *ppr278* plants and analyzed editing efficiency (T/(T + C)%). The T/(T + C) ratio was reduced among all editing sites in *atp8* and *nad3* in the *ppr278* mutant (Figure 5B and Appendix A). These results indicated that PPR278 is required for editing 94 C-to-U sites of mitochondrial transcripts related to mitochondrial complexes Ⅰ, Ⅳ, and Ⅴ, cytochrome c biogenesis, and ribosomal proteins (Appendix A).

To further investigate the effects of aberrant C-to-U RNA editing on protein translation, we investigated the coding sequence and corresponding amino acid sequences of all 35 mitochondrion-encoded transcripts in the *ppr278* mutant. Nearly 83.16% of C-to-U RNA editing events caused amino acid substitutions (Appendix A). Notably, the replacement of leucine accounted for more than 40% of all editing events, ranking highest in silenced and non-silenced sites (Appendix A) [35]. We also discovered that editing failed at 37 bases distributed among *atp1*, *atp4*, and *ccmF_N_* in the *ppr278-1* and *ppr278-2* mutants (Table 1). We analyzed the failed editing at these bases in the *ppr278* mutants and observed that editing at all bases except for *ccmF_N_*-417 and *atp1*-30 caused amino acid changes. The other failed editing sites in *ccmF_N_* resulted in amino acid substitutions at positions 11, 9, 4, 4, 3, 2, and 1, corresponding to Leu-to-Pro, Leu-to-Ser, Try-to-Arg, Phe-to-Ser, Ser-to-Pro, Cys-to-Arg, and Phe-to-Leu, respectively (Appendix A, Table 1). *ccmF* in *Escherichia coli*, the homolog of maize *ccmF_N_*, encodes a large protein with 11 transmembrane domains [52]. We used TMHMM (http://services.healthtech.dtu.dk/service.php?TMHMM-2.0) (Accessed on 9 March 2022) to analyze the transmembrane domains of *ccmF_N_* and determined that the lack of editing in *ccmF_N_* disrupted the 1st, 4th, 5th, and 6th transmembrane domains in the *ppr278* mutant (Appendix A). We also analyzed the amino acid conservation of C-to-U editing sites of *atp8*, *nad3*, and *ccmF_N_*, which indicated that these sites were relatively conserved in terms of amino acids change (Figure 5, Appendix A). In total, PPR278 is essential for C-to-U RNA editing in 10 mitochondrial transcripts, including 37 abolished sites in *atp1*, *atp4*, and *ccmF_N_*. Loss of function of PPR278 lead to a series of amino acid substitution events due to failed C-to-U RNA editing. 

### 2.6. Mutation of PPR278 Alters the Expression of Mitochondrial Function-Related Genes

Considering the nuclear localization of PPR278 and its implication in nuclear gene regulation, we performed RNA-seq analysis using 18-DAP endosperms and embryos from *ppr278-1* and WT plants. The materials were collected from segregating ears with two biological replicates (Figure 6A). Those with a fold change and *p*-value above the threshold (fold change > 2 and *p*-value < 0.05) were identified as differentially expressed genes (DEGs). We identified 2466 DEGs between *ppr278-1* and WT plants, including 1542 up-regulated DEGs and 924 down-regulated DEGs (Figure 6B, Appendix A).

Gene Ontology (GO; http://systemsbiology.cau.edu.cn/agriGOv2/) (Accessed on 9 March 2022) analysis using the DEGs indicated that both down- and up-regulated DEGs were enriched in the term ‘post-embryonic development’ (GO: 0009791). The reason why this Go term was enriched might be interpreted as genes transcribed actively to regulate post-embryonic development rather than direct results of loss-function-of *PPR278*. The other down-regulated DEGs were mostly associated with energy-consuming processes, such as ‘metabolic process’, ‘nutrient reservoir activity’, ‘plant and seed development’, and ‘nucleosome assembly’ [12,13], suggesting that energy production in the *ppr278-1* mutant is insufficient to maintain plant growth and seed development. The remaining up-regulated DEGs were closely related to ‘mitochondrial part’ (GO: 0044429), which is predicted to function in mitochondria. For example, ‘electron transport chain’ (GO:0022900), ‘mitochondrial respiratory chain complex Ⅰ’ (GO:0005747), ‘II’ (GO: 005749), and ‘III’ (GO:0005750), ‘ATP synthase activity’ (GO:0000276, GO:0042775, GO:0005753), ‘mitochondrial respiratory chain complex assembly’ (GO:0033108), ‘precatalytic spliceosome’ (GO: 0071011), ‘U1 small nuclear ribonucleoprotein particle (snRNP)’(GO:0005685), ‘U2 snRNP’ (GO:0005686), ‘U4/U6×U5 tri-snRNP complex’ (GO:0046540), and ‘pre-catalytic spliceosome’ (GO:0071011) (Figure 6C, Appendix A) were enriched in *ppr278*. These findings demonstrated that the *ppr278* mutation significantly altered the expression of genes putatively involved in mitochondrial processes, leading to impaired mitochondrial function and retarded seed development.

## 3. Discussion

### 3.1. PPR278 Functions in RNA Cis-Splicing and RNA Editing

P-subfamily PPR proteins are generally involved in RNA splicing [4]. However, three PPR proteins of the P-subfamily function in C-to-U RNA editing in *Arabidopsis*. For example, PPR596 influences editing efficiency at partial editing sites (rps3eU64RW, rps3eU603FF, rps3eU887SL, rps3eU1344SS, rps3eU1352PL, rps3eU1470SS, rps3eU1534RC, rps3eU1571AV, rps3eU1580SF, rps3eU1598SL, Pseudo-rps14eU99, and Pseudo-rps14Eu194) in mitochondrial transcript *rps3* and *rps14*, while another gene P-type protein, PPR-MODULATING EDITING (PPME), is responsible for C-to-U RNA editing at both sites *nad1*-898 and *nad1*-937 [53,54,55]. Here, we characterized PPR278 as a splicing factor that participates in intron splicing of *nad2* intron 4 and *nad5* introns 1 and 4. In addition, PPR278 was also recognized as an editing factor that is responsible for 94 sites of C-to-U RNA editing in mitochondrial transcripts.

PPR278 is necessary for intron splicing of *nad2* intron 4 and *nad5* introns 1 and 4, which further highlights the role of P-subfamily PPR proteins in intron splicing. Multiple PPR proteins and other splicing factors may participate in splicing the same intron. For instance, *nad2* intron 4 cis-splicing was also abnormal in *emp16* and *emp12* mutants in maize [9,17]. ZmSMK9 is involved in the cis-splicing of *nad5* introns 1 and 4 [18]. Furthermore, EMP25, PPR101, and PPR231 are required for splicing *nad5* intron 1 in maize. Thus, mutations in these PPR proteins affect the cis-splicing of mitochondrial *nad5* introns [18,21,22]. Similarly, EMP11, DEK2, and EMP8 are associated with nad1 intron splicing [12,13,14,15,16]. In addition, EMP10 and DEK37 are necessary for cis-splicing of *nad2* intron 1 [11,56]. These findings indicate that, in its function as a spicing factor, PPR278 may function in spliceosome complexes in mitochondria, possibly in complexes with these other factors that also affect splicing of the same introns.

In addition to its splicing role–and unlike previously reported P-type PPR proteins–PPR278 functions in C-to-U RNA editing in genes associated with other mitochondrial complexes besides complex III (Appendix A), particularly in *ccmF_N_* with 30 abolished C-to-U RNA editing sites in the *ppr278* mutant (Appendix A, Table 1). Failed editing of *ccmF_N_* abolished the formation of several transmembrane domains (Appendix A), especially the formation of the 5th transmembrane domain, which is important for protein conformation [32]. *ccmF_N_*-302 and *ccmF_N_*-1553 are essential for C-to-U editing and the biogenesis of cytochrome c (Appendix A) [31,32,36]. Additionally, a conserved WWD domain (G [2X] W [2 or 4X] WG [2X] WXWD) consisting of two W-to-R substitutions in sites *ccmF_N_*-1357 and *ccmF_N_*-1375 is required for accessibility for heme b attachment to cytochromes c1 and c [26,32]. Another conserved histidine (H) in site *ccmF_N_*-1342, called PHis2 involved in heme binding, was also fully unedited in the *ppr278* mutant [57,58] (Appendix A). Moreover, *atp1*-1292 and *atp4*-59 sites are important for the assembly of mitochondrial complex Ⅴ [31,32,35,36]. Thus, loss of PPR278 function led to 37 abolished C-to-U RNA editing sites and an abnormal mitochondrial complex in maize (Table 1).

In general, P-subfamily members primarily function in splicing and PLS-subfamily members primarily function in editing. PPR278 is the first P-subfamily member determined to be involved in intron splicing of mitochondrial transcripts and C-to-U RNA editing in maize. However, several other PPR proteins act simultaneously as splicing and editing factors in plants. The E-type PPR protein DEK55 plays a crucial role in RNA editing at multiple sites as well as in the splicing of *nad1* and *nad4* introns in maize, and the DYW-type PPR protein BLX (Baili Xi) controlling mitochondrial RNA editing and splicing is essential for seed development in *Arabidopsis* [15,16,19,59]. These discoveries have expanded our knowledge about PPR proteins: The function of a PPR protein is not dependent on its classification category. There is a distinguishable divergence function in P-subfamily and PLS-subfamily members, and PPR278 clustered in the P-family by phylogenetic analysis (Appendix A). The irregular motif *p* and classification of PPR278 provide a plausible explanation for why PPR278 has both splicing and editing functions in mitochondria. Notably, no C-to-U editing sites were detected in *nad2* and *nad5* cDNA sequences in the *ppr278* mutant, suggesting that *nad2* intron 4 and *nad5* introns 1 and 4 have no effect on the editing of the exon sequence. This implied that PPR278 might conduct C-to-U RNA editing and intron splicing independently [9]. 

### 3.2. PPR278 Is Involved in Regulation of Nuclear Gene Expression

Several nucleus-localized PPR proteins are involved in gene regulation [4,5,42]. These nucleus-localized PPR proteins (SOAR1, GRP23, PNM1, OsNPPR1, and OsNPPR3) interact with transcription factors and/or RNA polymerases to regulate the transcription and/or mRNA processing of nuclear genes associated with mitochondrial function or localized to mitochondria, and likely function as potential regulators of gene expression during embryogenesis in plants [43,44,45,46,47]. In maize, DEK43 targets to nuclei, but its functions in nuclei remain largely unknown [19]. In our study, PPR278 localized to nuclei (Figure 3A,B), and RNA-seq analysis showed that the expression of many genes was significantly changed in the *ppr278-1* mutant (Figure 6B). 

Genes related to post-embryonic development were dramatically down- and up-regulated in *ppr278* mutants (Figure 6C). Moreover, genes associated with plant and seed development were widely down-regulated in the *ppr278-1* mutant. This down-regulation might explain the lower germination rate and seedling-lethal phenotype of the *ppr278* mutant. Notably, GO terms GO:0006334 (nucleosome assembly essential for a series of energy-consuming biological processes [60]) and GO:0045735 (nutrient reservoir activity associated with kernel development and filling) were also significantly enriched in the *dek2*, *dek10*, *dek35*, *dek37*, and *dek41* mutants, characterized by defective kernels and abnormal mitochondrial function [10,11,12,13,61]. The down-regulated genes in *ppr278-1* plants were enriched for both kinds of these GO terms. In addition, 7 and 23 DEGs were highly associated with the GO terms ‘spliceosome’ and ‘mitochondrial respiratory chain’, respectively, and exhibited extensive up-regulation (Figure 6C, Appendix A). This enrichment suggests that the mitochondrial oxidation respiratory chain was severely disrupted in *ppr278* mutants. It remains to be tested whether the altered expression of nucleus-encoded genes arises from PPR278 binding their transcripts directly in the nucleus or if it is a consequence of arrested mitochondrial function. Based on our results, we conclude that PPR278 has a precise regulatory role in the expression of mitochondrial and nuclear genes and might function in transcription and processing of nuclear mRNA.

## 4. Materials and Methods

### 4.1. Plant Materials

The maize (*Zea mays* L.) *ppr278-1* and *ppr278-2* mutants were obtained from our EMS-based mutant library. In brief, the mature pollen of inbred line B73 was treated with 0.015% ethyl methanesulfonate solution (vol/vol) in a 1:10 pollen:solution ratio for 30 min. The resulting M1 plants were self-pollinated, and M2 seeds showing a Mendelian segregation ratio were selected. M2 plants were further self-pollinated to obtain M3 progenies for bulked pools. The 12-DAP embryos and endosperms from 20 WT and 20 mutant-type kernels from segregating M3 ears were used for bulked segregant analysis. The plants were grown in the experimental field at Shangzhuang Experimental Base of China Agricultural University under natural conditions (Beijing, China).

### 4.2. Bulked Segregant Analysis

The 12-DAP embryos and endosperms samples were ground in liquid nitrogen, and total RNA was extracted with TRIzol Reagent (Thermo Fisher SCIENTIFIC, 15596018, Waltham, MA, USA) according to the manufacturer’s instructions. The cDNA library was prepared with VAHTS Stranded mRNA-seq Library Prep Kit for Illumina V2 (Vazyme, NR612, Nanjing, China) and sequenced on the Illumina X-ten platform. Raw reads were aligned to the B73 reference genome (V4) using HISAT2. The alignment files were converted to BAM format and sorted using Samtools. SNP calling, including C-to-T and G-to-A mutations, was performed by Samtools and Bcftools using unique reads. High-confidence SNPs were selected using the following criteria: (1) the SNP was supported by at least five reads, and (2) the SNP was detected in both the mutant pool and its corresponding WT pool. Functional annotation of SNPs was performed using SnpEff to screen SNPs between the two RNA pools. The SNP index was listed in Appendix A.

### 4.3. Candidate Gene Analysis and Validation

We identified 14 genes with missense mutations, 10 genes with synonymous mutations, 5 genes with 3ʹUTR mutations, 2 genes with 5ʹUTR mutations, and 1 gene with stop codon-gained mutation (*Zm00001d015156*) in WT and mutant pools from our bulked segregant analysis. Then the ΔSNP index (SNP index difference between the mutant bulk and WT for each SNP) was calculated for each SNP. *Zm00001d015156* was considered as candidate gene because of its mutant type and the highest SNP index among these genes. To test the degree of linkage between the candidate gene *Zm00001d015156* and the mutant phenotype in the M3 segregating population, we designed primers PPR278-1F/R and PPR278-2F/R closely linked with the mutant sites (Appendix A). Next, we used heterozygous *ppr278-1* and *ppr278-2* plants for reciprocal pollination followed by genetic analysis using the kernels from the pollinated ears (Figure 2B). 

### 4.4. Subcellular Localization of PPR278

The full-length PPR278 coding sequence without the stop codon was inserted into the vectors pSuper1300::eGFP and pSPYNE-35S::eGFP. Approximately 1 µg of the recombinant plasmid PPR278::GFP was transformed into onion (*Allium cepa* L.) epidermal cells and maize protoplasts. After culture for 20 h at 28 °C, the fluorescence signals were detected by a LSM710 confocal laser scanning microscope (Carl Zeiss, Oberkochen, Germany). AtAHL22-mCherry was used as a nuclear marker [62]. 

### 4.5. Phylogenetic Analysis 

Sequences of PPR278 and its orthologous proteins were downloaded from either NCBI (Bethesda, MD, USA) or PPR energy databases (Perth, Australia) [49,50]. The full-length protein sequences were used to construct a phylogenetic tree with the neighbor-joining method using Mega-X software.

### 4.6. RT-PCR and qRT-PCR

Total RNA was treated with RQ1-free DNase I (Promega, M610A, Madison, WI, USA) to completely remove DNA contamination. The cDNA was synthesized using oligo (dT) primers (Promega, C1181, Madison, WI, USA). Quantitative reverse transcription PCR (qRT-PCR) was performed using TB Green Premix Ex Taq (Takara, RR420A, Toyobo, Osaka, Japan) with an ABI 7500 system (Thermo Fisher SCIENTIFIC, Waltham, MA, USA), and relative gene expression was calculated with the 2^−^^ΔΔCt^ method as described previously [63]. The data were normalized using *ZmActin* as a control. The experiments were replicated two times, and the primers used in the qRT-PCR (PPR278-764F and PPR278-1003R) are listed in Appendix A. 

### 4.7. Analysis of RNA Splicing by PPR278

Total RNA was extracted and reverse transcribed with random primers as described above. The expression differences of maize mitochondrial transcripts between WT and *ppr278-1* plants were compared by PCR using *18s RNA* as the reference gene. The primers used for mitochondrial genes were described previously [31]. The expression levels in WT and *ppr278-1* plants were normalized using *ZmActin*. The PCR reaction buffer and DNA polymerase were from Mei5bio (MF743-10, Beijing, China). To further analyze RNA splicing, we chose the primers covering group II introns as described previously [9].

### 4.8. Analysis of RNA Editing by PPR278

Mitochondrial RNA was reverse transcribed with random primers as described above. The full-length sequences of 35 mitochondrial transcripts in *ppr278-1* and *ppr278-2* kernels at 12-DAP were reverse transcribed and directly sequenced as described previously [31]. The C-to-U RNA editing status of mitochondrial transcripts from WT and *ppr278-1* plants was compared. The editing sites identified in the *ppr278-1* mutant were confirmed in the *ppr278-2* mutant. The editing efficiency of *atp8* and *nad3* in WT, *ppr278-1*, and *ppr278-2* plants was estimated by cloning the *atp8* and *nad3* coding sequences into vector B-zero (TransGen Biotech pEASY-Blunt Zero Cloning Kit, CB501-01, Beijing, China) and sequencing more than 40 independent colonies for each gene. 

### 4.9. RNA-Sequencing Analysis

Total RNA of 18-DAP endosperm and embryo (4 µg) tissues was extracted as described above, and two *ppr278-1* or WT biological samples were collected for construction of the RNA-sequencing library. The cDNA library was performed according to Hieff NGS MaxUp II Dual-mode mRNA Library Prep Kit for Illumina (Yeasen, H9001360, Beijing, China) and sequenced by the Illumina NovaSeq platform for 150-nucleotide paired-end reads. Raw reads were aligned to the B73 reference genome (V4) using HISAT2. Data were normalized as fragments per kilobase of exon per million fragments mapped (FPKM), since the sensitivity of RNA-seq depends on the transcript length. The significant DEGs were filtered with the criterion: fold change > 2 and *p*-value < 0.05.

## Figures and Tables

**Figure 1 ijms-23-03035-f001:**
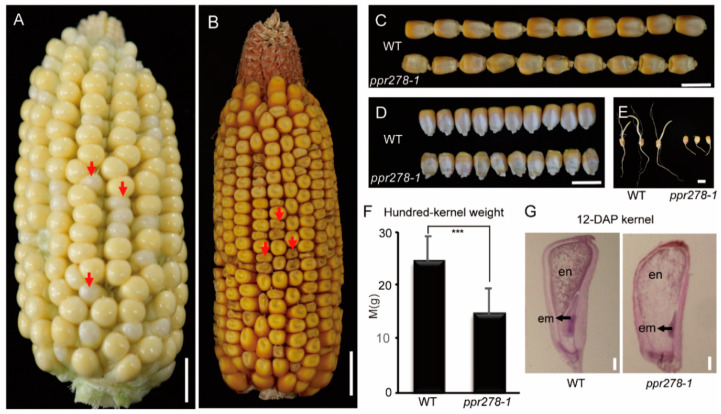
*ppr278-1* mutants exhibited defective kernel development and lethal seedlings. (**A**,**B**) Ears of self-pollinated heterozygous *ppr278-1* plants at 16 days after pollination (16 DAP) (**A**) and at maturity (**B**). Red arrows indicate the mutant kernels. Bar = 1 cm. (**C**,**D**). Comparison of the length (**C**) and width (**D**) of mature seeds of the WT and the *ppr278-1* mutant. Bar = 1 cm. (**E**) Germination of WT and *ppr278-1* mutant seeds after dark culture for two days in an incubator. Bar = 1 cm. (**F**) Comparison of the 100-kernel weight of fully-mature WT and *ppr278-1* mutant kernels. Values and error bars represent the mean and standard deviation of three biological replicates. *** *p* < 0.001 (*t*-test), representing a significant difference. (**G**) Longitudinal section of WT (**left**) and *ppr278-1* mutant kernels (**right**) at 12-DAP. en: endosperm, em: embryo. Bar = 0.5 mm.

**Figure 2 ijms-23-03035-f002:**
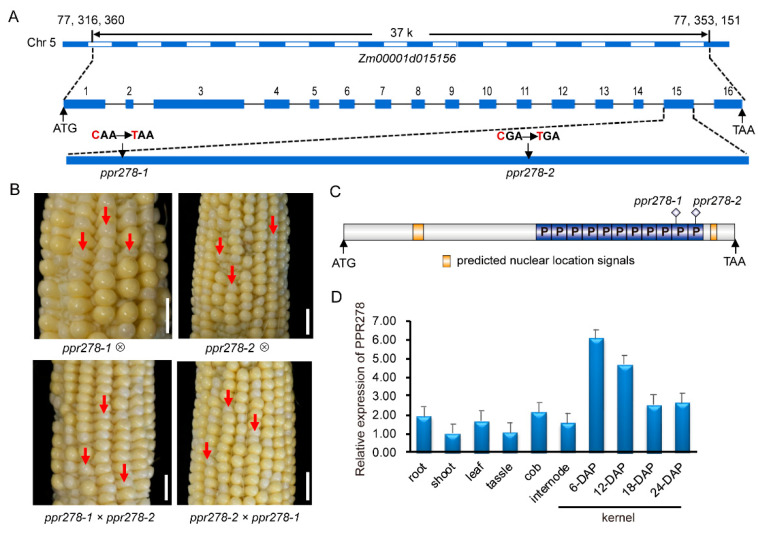
Identification of the *ppr278* mutation. (**A**) Schematic of the *Zm00001d015156* gene and the location of mutated sites in the *ppr278-1* and *ppr278-2* mutants. The candidate gene was located on chromosome 5 and identified as *Zm00001d015156*, which contains 16 exons (blue rectangles) and 15 introns (black lines). A C-to-T mutation in the 15th exon leads to protein termination at positions 2269 and 2404 bp from the ATG start codon in the *ppr278-1* and *ppr278-2* mutants, respectively. (**B**) The results of the allelic test of 12-DAP ears of the self-pollinated plants of PPR278/*ppr278-1*, PPR278/*ppr278-2*, PPR278/*ppr278-1* ×PPR278/*ppr278-2*, and PPR278/*ppr278-2*×PPR278/*ppr278-1*. Red arrows indicate mutant *ppr278* seeds. Bar = 1 cm. (**C**) Schematic of PPR278 protein. Blue rectangles represent the *p* motifs. (**D**) qRT-PCR analysis of relative PPR278 expression in different tissues and kernels at different developmental stages.

**Figure 3 ijms-23-03035-f003:**
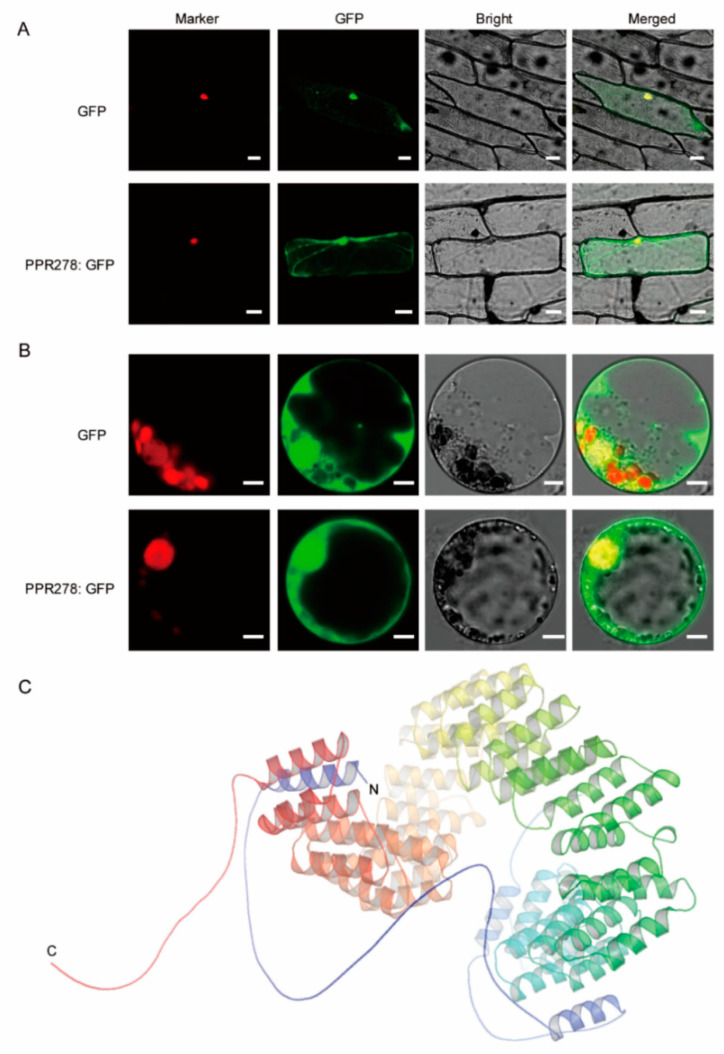
PPR278 was localized to the nucleus and mitochondria and predicted to form a series of alpha helices. (**A**,**B**) Transient expression of PPR278 in onion epidermal cells (**A**) and maize protoplasts (**B**). GFP was used as a control (top panels of (**A**,**B**)) and was co-expressed with the nuclear marker (bottom panels of (**A**,**B**)). Bar = 50 μm (**A**) and 5 μm (**B**). (**C**) Predicted protein structure of PPR278. The purple and red helices indicate the N terminus (N) and C terminus (C) of PPR278, respectively.

**Figure 4 ijms-23-03035-f004:**
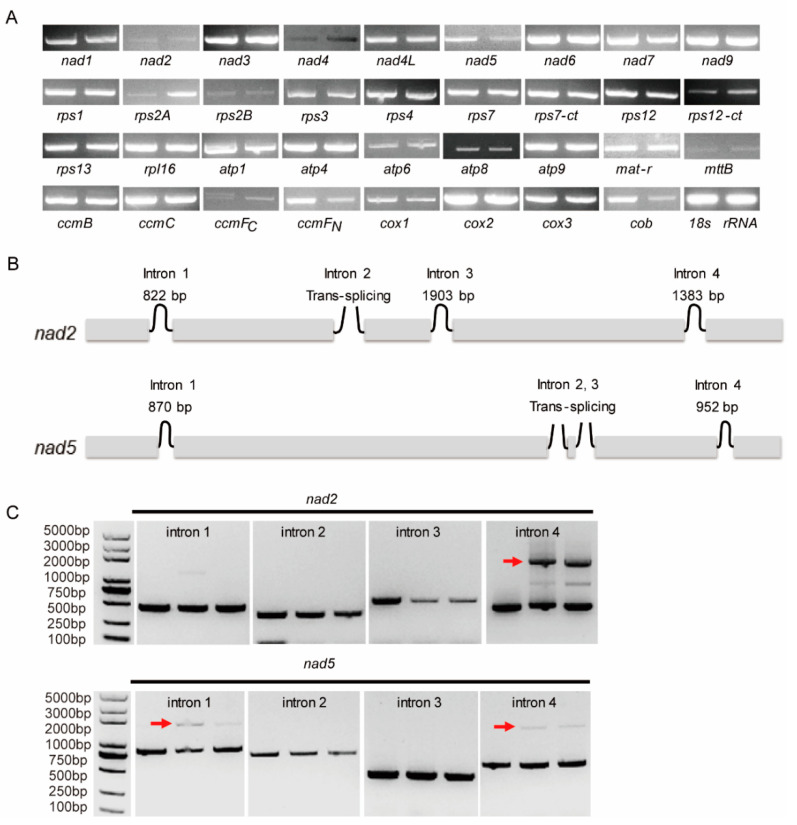
Intron splicing deficiency of *nad2* and *nad5* in *ppr278* mutants. (**A**) RT-PCR analysis of 35 mitochondrion-encoded transcripts in WT (left) and *ppr278-1* (right) plants. RNA was isolated from WT and *ppr278-1* endosperm from the same segregating ear at 12-DAP. *18S rRNA* served as an internal control. (**B**) Structures of the maize mitochondrial genes *nad2* and *nad5*. (**C**) RT-PCR analysis of *nad2* and *nad5* intron splicing efficiency in WT (left), *ppr278-1* (middle), and *ppr278-2* (right) plants. A DNA marker is shown in the farther left lane. Red arrows indicate abnormal splicing in the *ppr278-1* and *ppr278-2* mutants.

**Figure 5 ijms-23-03035-f005:**
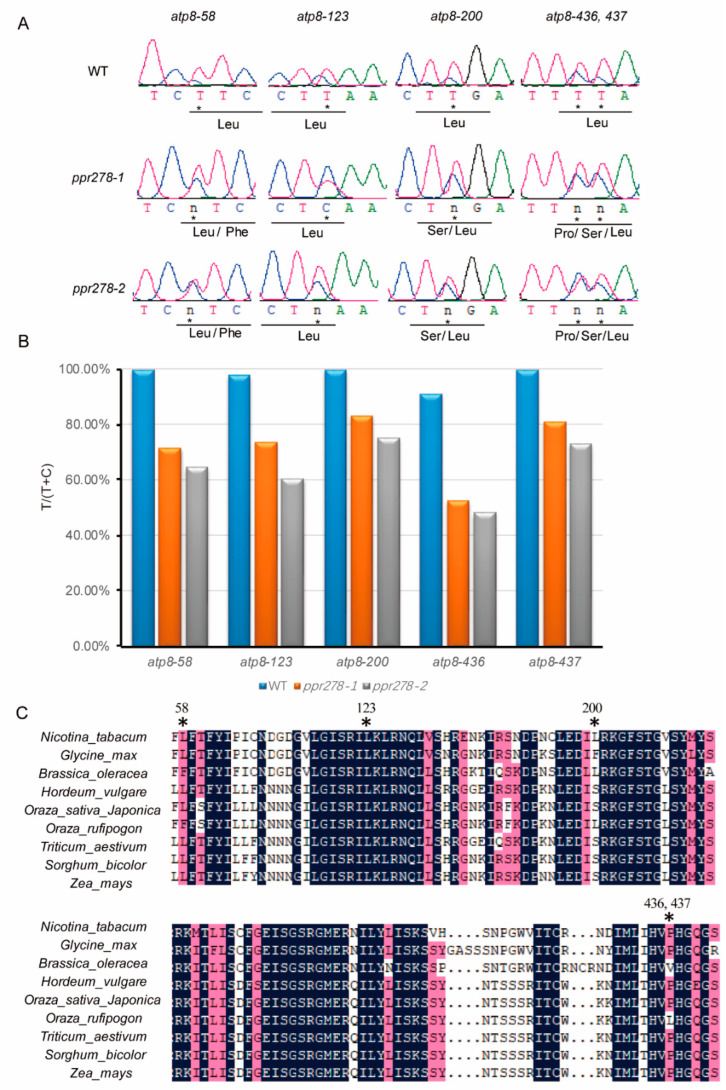
C-to-U RNA editing sites and efficiency in *atp8* transcripts. (**A**) Editing profile of five C-to-U RNA editing sites in the *atp8* transcripts in the *ppr278* mutant. (**B**) C-to-U editing efficiency in *atp8* transcripts; “n” stands for the peak of heterozygous genotype (C/U) in the same editing site, “*” stands for the C-to-U site in *atp**8*. (**C**). Comparison of amino acid sequences of *atp8* in different species.

**Figure 6 ijms-23-03035-f006:**
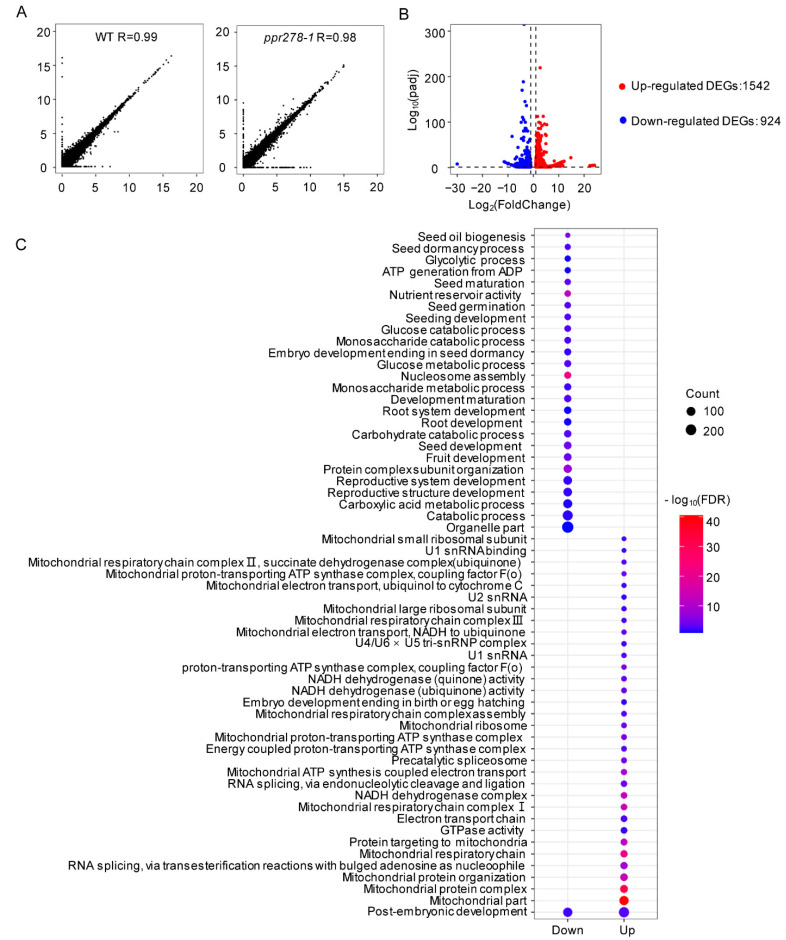
Transcriptome analysis in the *ppr278-1* mutant. (**A**) Correlation between the replicates of RNA-seq data in WT and *ppr278-1* plants. The x- and y-axes represent replicate 1 and replicate 2, respectively. (**B**) Volcano plot displaying differentially expressed genes (DEGs) between WT and *ppr278-1* plants. The *y*-axis corresponds to the mean expression value of log10 (*p*-value), and the *x*-axis displays the log2 (fold change) value. Red dots represent the up-regulated transcripts between WT and *ppr278-1* plants. Blue dots represent the down-regulated transcripts. (**C**) GO terms of the top 33 and 27 up- and down-regulated DEGs, respectively, in WT and *ppr278-1* plants. The size of the dot represents the number of genes. The color shows the value of −Log10 (FDR, false discovery rate).

**Table 1 ijms-23-03035-t001:** Comparison of editing profiles that differ between WT and *ppr278*.

Gene	Position from ATG	C-to-U Editing Efficiency	WT	*ppr* *278-1*	*ppr* *278-2*	Gene	Position from ATG	C-to-U Editing Efficiency	WT	*ppr* *278-1*	*ppr* *278-2*
*atp1*	30	A	Leu	Leu	Leu	*nad3*	275	D	Ser/Phe	Ser/Phe	Ser/Phe
*atp1*	1178	A	Leu/Ser	Ser	Ser	*nad3*	317	D	Ser/Phe	Ser/Phe	Ser/Phe
*atp1*	1292	A	Leu/Pro	Pro	Pro	*nad3*	344	D	Ser/Leu	Ser/Leu	Ser/Leu
*atp1*	1490	A	Leu/Pro	Pro	Pro	*nad3*	349	D	Arg/Trp	Arg/Trp	Arg/Trp
*atp1*	1499	A	Phe/Ser	Ser	Ser	*rpl16*	228	D	Ile	Ile	Ile
*atp4*	56	A	Leu	Pro	Pro	*rps2A*	200	D	Ser/Phe	Ser/Phe	Ser/Phe
*atp4*	59	A	Phe	Ser	Ser	*rps2A*	449	D	Ala/Val	Ala/Val	Ala/Val
*atp4*	71	D	Leu	Ser	Leu/Ser	*rps2A*	514	D	Ser	Ser/Pro	Ser/Pro
*atp4*	76	D	Ser	Pro	Pro	*rps2A*	541	D	Cys	Cys/Arg	Cys/Arg
*atp4*	89	D	Leu	Ser	Leu/Ser	*rps2A*	548	D	Leu	Ser/Leu	Ser
*atp4*	118	D	Cys	Arg	Cys/Arg	*rps2B*	550	I	Arg/Cys	Cys	Cys
*atp4*	360	D	Cys	Cys	Cys	*rps12*	71	D	Ser/Leu	Ser/Leu	Ser/Leu
*atp4*	407	D	Leu	Ser	Leu/Ser	*rps12*	196	D	His/Tyr	His/Tyr	His/Tyr
*atp4*	428	D	Ile	Thr	Ile/Thr	*rps12*	221	D	Ser/Leu	Ser/Leu	Ser/Leu
*atp8*	58	D	Leu/Phe	Leu/Phe	Leu/Phe	*rps12*	269	D	Ser/Leu	Ser/Leu	Ser/Leu
*atp8*	123	D	Leu	Leu	Leu	*rps12*	284	D	Ser/Leu	Ser/Leu	Ser/Leu
*atp8*	200	D	Ser/Leu	Ser/Leu	Ser/Leu	*rps12*	289	D	Arg/Cys	Arg/Cys	Arg/Cys
*atp8*	436	D	Pro/Ser /Leu	Pro/Ser /Leu	Pro/Ser /Leu	*ccmF_N_*	76	A	Ser	Pro	Pro
*atp8*	437					*ccmF_N_*	77				
*cox3*	69	D	Leu	Leu	Leu	*ccmF_N_*	137	A	Leu	Pro	Pro
*cox3*	245	D	Leu/Pro	Leu/Pro	Leu/Pro	*ccmF_N_*	176	A	Leu	Ser	Ser
*cox3*	257	D	Phe/Ser	Phe/Ser	Phe/Ser	*ccmF_N_*	181	A	Cys	Arg	Arg
*cox3*	289	D	Phe/Ser	Phe/Ser	Phe/Ser	*ccmF_N_*	190	A	Ser	Pro	Pro
*cox3*	311	D	Ser/Phe	Ser/Phe	Ser/Phe	*ccmF_N_*	287	A	Leu	Ser	Ser
*cox3*	314	D	Phe	Ser	Ser/Phe	*ccmF_N_*	302	A	Leu	Pro	Pro
*cox3*	413	D	Leu	Pro	Leu/Pro	*ccmF_N_*	401	A	Phe	Ser	Ser
*cox3*	422	D	Leu	Pro	Leu/Pro	*ccmF_N_*	410	A	Leu	Ser	Ser
*cox3*	527	D	Phe	Ser/Phe	Ser/Phe	*ccmF_N_*	417	A	Phe	Phe	Phe
*cox3*	566	D	Phe/Ser	Ser	Phe/Ser	*ccmF_N_*	743	A	Leu	Pro	Pro
*cox3*	754	D	Trp	Arg	Trp/Arg	*ccmF_N_*	752	A	Leu	Ser	Ser
*nad3*	5	D	Ser/Leu	Ser/Leu	Ser/Leu	*ccmF_N_*	790	A	Cys	Arg	Arg
*nad3*	39	D	Ile	Ile	Ile	*ccmF_N_*	812	A	Leu	Ser	Ser
*nad3*	44	D	Pro/Leu	Pro/Leu	Pro/Leu	*ccmF_N_*	824	A	Leu	Pro	Pro
*nad3*	61	D	Pro/Ser /Leu	Pro/Ser /Leu	Pro/Ser /Leu	*ccmF_N_*	839	A	Leu	Ser	Ser
*nad3*	62					*ccmF_N_*	1325	A	Leu	Pro	Pro
*nad3*	80	D	Pro/Leu	Pro/Leu	Pro/Leu	*ccmF_N_*	1342	A	Tyr	His	His
*nad3*	137	D	Ser/Phe	Ser/Phe	Ser/Phe	*ccmF_N_*	1357	A	Trp	Arg	Arg
*nad3*	138	D				*ccmF_N_*	1375	A	Trp	Arg	Arg
*nad3*	146	D	Ser/Phe	Ser/Phe	Ser/Phe	*ccmF_N_*	1408	A	Trp	Arg	Arg
*nad3*	185	D	Pro/Leu	Pro/Leu	Pro/Leu	*ccmF_N_*	1469	A	Leu	Ser	Ser
*nad3*	190	D	Pro/Ser	Pro/Ser	Pro/Ser	*ccmF_N_*	1489	A	Phe	Leu	Leu
*nad3*	208	D	Pro/Ser /Leu	Pro/Ser /Leu	Pro/Ser /Leu	*ccmF_N_*	1493	A	Leu	Pro	Pro
*nad3*	209					*ccmF_N_*	1505	A	Leu	Ser	Ser
*nad3*	215	D	Pro/Leu	Pro/Leu	Pro/Leu	*ccmF_N_*	1540	A	Ser	Pro	Pro
*nad3*	230	D	Ser/Phe	Ser/Phe	Ser/Phe	*ccmF_N_*	1553	A	Phe	Ser	Ser
*nad3*	247	D	Pro/Ser	Pro/Ser	Pro/Ser	*ccmF_N_*	1588	A	Trp	Arg	Arg
*nad3*	251	D	Pro/Leu	Pro/Leu	Pro/Leu	*ccmF_N_*	1709	A	Leu	Pro	Pro

The C-to-U editing efficiency of mitochondrial transcripts in *ppr278-1* and *ppr278-2* mutants was abolished (A), decreased (D), or increased (I) when compared with the WT. Leu: Leucine, Ser: Serine, Pro: Proline, Phe: Phenylalanine, Cys: Cysteine, Arg: Arginine, Ile: Isoleucine, Thr: Threonine, Try: Tryptophan, Ala: Alanine, Val: Valine, His: Histidine, Tyr: Tyrosine.

## Data Availability

Not applicable.

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
