# Peer review of "Maize PPR278 Functions in Mitochondrial RNA Splicing and Editing"

_ijms, 2022, doi:10.3390/ijms23063035_

Round 1

Reviewer 1 Report

Yang et al. explored the biological relevance of the maize pentatricopeptide repeat protein, PPR278, in RNA metabolism. Overall, the experimental design was straightforward and the results were depicted logically along with decent English.

My major concern is the subcellular localization of PPR278. By employing the AHL22-mCherry as a marker for the nuclear-localization, the authors showed a partial overlapping pattern of PPR278-GFP with a nuclear marker in both onion epidermis and maize protoplast cells (Figure 3). However, no evidence supporting mitochondrial localization of PPR278 proteins was provided in this study. Without such a shred of evidence, I find it inappropriate to state that PPR278 protein is a dual-localized protein, although I am fairly certain from the results shown in Figures 4 and 5 that intron splicing and RNA editing are influenced by PPR278 proteins in the mitochondria. My suggestion is to do a co-localization analysis of PPR278 with the mitochondria marker. Alternatively, refrain from mentioning mitochondria localization of PPR278 proteins.

Minor points:

(1) RNA-seq results; I am not surprised at the fact that altered genes are enriched in the GO category “post-embryonic development”. This does not necessarily mean that PPR278 protein (complex) preferentially targets RNA molecules whose translated proteins function in post-embryonic development. Rather, I interpret such result as a result of the RNA source tissues, that said, “endosperm and embryo at 18 DAP” in which genes regulating the post-embryonic development are actively transcribed and hence an altered gene expression between wild-type and the mutant are highlighted. I suggest authors consider mentioning this interpretation as one of the alternative ways to account for differential nuclear gene expression.

(2) Up-regulated genes in RNA-seq data, as well as semi-quantitative RT-PCR of mitochondria genes (Figure 4A); Are they up-regulated as a negative-feedback regulation?

(3) Lines 305 and 329; nad2 is not italic. Please make sure if this is correct.

(4) Line 198; Does “abnormal splicing” mean intron retention?

(5) Figure 4B and 4C; As for nad2 intron2, I am wondering if the authors could detect trans-splicing defects by RNA-seq, rather than intron retention… Or is nad2 intron 2 regulated by intron retention, as well? The same principle applies to introns 2 and 3 of nad5.

(6) Figure 4C. It seems better if the red arrow is rotated 90 or 45 degrees (clockwise) so that only the higher band (intron retention?) is indicated by the arrow. DNA markers are provided, however, it would be helpful if the actual size is indicated so that the reader can grasp the size of PCR products and the gap in between two bands if any.

(7) Figure 4C; The transcript abundance of intron 3 flanking region of nad2 in ppr278 appears to be lower than that of wild-type and the size of intron 3 is the biggest. Does it imply that the intron 3 retained transcripts failed to amplify by the PCR conditions the authors examined??? Or is there another interpretation to account for the reduced nad2 transcripts only around the intron 3 region?

(8) Subsection "4.3 Candidate gene analysis and validation in Material and method" (lines 393-400). The description is casual. Please elaborate on it to make it a bit conventional.

(9) Table 1; please make sure that the position of “S/F” in nad3 at position 137 in ppr278-2 is correct.

(10) Figure 4A, 4C, Figure S7; Were primer names and sequences used for RT-PCR provided in the submitted MS?

  1.  

Author Response

3 March, 2022

Dear Ms. Jelena Romcevic,

Thank you for reviewing out manuscript and giving us the opportunity to revise our manuscript. Please find our revised manuscript entitled “Maize PPR278 localizing to the nucleus and mitochondria functions in mitochondrial RNA splicing and editing” with the manuscript ID ijms-1587245 in the accompanying file.

Meanwhile, we also appreciate the reviewers for their thoughtful comments, which are very helpful for us to revise and improve our manuscript significantly. According to their suggestions, we have addressed all referee comments in the document “Reply to the Review report (Reviewer1)”. Please see the attachment. The font color of referee comments is black, while the font color of our responses is red; besides, line and page numbers refer to the revised manuscript.

We look forward to hearing from you. Please feel free to contact us if you require any additional information.

Yours sincerely,

Weibin Song, Ph.D

Professor of Plant Genetics and Breeding

National Maize Improvement Center

China Agricultural University

Phone: 86-10-62731405

Fax: 86-10-62731416

E-mail: songweibin@cau.edu.cn

Reviewer 2 Report

Please, take into consideration the following remarks when you revise your manuscript:

Line 2: Please, re-phrase the title: it must be “Maize PPR278 localization”; also, do localization of PPR278 to the nucleus has an effect on mitochondrial RNA splicing and editing? It looks so by your title, which may be misleading.

Line 14: “Land plants” is informal. Be more specific and scientifically correct.

Line 19-20: PPR278 was expressed in all investigated tissues.

Line 30 and everywhere else: Here, and in all Latin names of plant species use the correct abbreviation for the author’s name, e.g., Zea mays L., Arabidopsis thaliana L. etc.

Line 202-206 and everywhere else: Be consistent when abbreviating amino acids – use either one or three letter code and not a mix of both.

Author Response

3 March, 2022

Dear Ms. Jelena Romcevic,

Thank you for reviewing out manuscript and giving us the opportunity to revise our manuscript. Please find our revised manuscript entitled “Maize PPR278 localizing to the nucleus and mitochondria functions in mitochondrial RNA splicing and editing” with the manuscript ID ijms-1587245 in the accompanying file.

Meanwhile, we also appreciate the reviewers for their thoughtful comments, which are very helpful for us to revise and improve our manuscript significantly. According to their suggestions, we have addressed all referee comments in the document “Reply to the Review report (Reviewer2)”. Please see the attachment. The font color of referee comments is black, while the font color of our responses is red; besides, line and page numbers refer to the revised manuscript.

We look forward to hearing from you. Please feel free to contact us if you require any additional information.

Yours sincerely,

Weibin Song, Ph.D

Professor of Plant Genetics and Breeding

National Maize Improvement Center

China Agricultural University

Phone: 86-10-62731405

Fax: 86-10-62731416

E-mail: songweibin@cau.edu.cn

Round 2

Reviewer 1 Report

Dear authors,

Thank you for considering my comments on the original manuscript into consideration. The quality of the resubmitted manuscript is improved and I find it worth publication in the International Journal of Molecular Sciences.

I have two comments on the revised manuscript.

(1) Lines 261-263

According to my suggestion, the authors provided a new sentence. This is the result section where interpreting the obtained results is not mandatory. I suggest authors consider replacing the sentence highlighted in yellow with the following.

This is consistent with the fact that the RNA-seq data was derived from the embryo-related tissues.

(2) Authors tried to elaborate on PPR278 localization in mitochondria that resulted in poor colocalization with mitochondria marker. I suggest the authors mention it in the discussion at the end of the first paragraph (line 300) by adding the following sentences if you will. This is my humble opinion, not mandatory.

In this study, we exerted our efforts on elucidating the subcellular localization of PPR278 and found the preferential localization in the nucleus and cytoplasm. However, despite the fact that the PPR278-GFP construct was driven by CaMV 35S promoter, the fluorescence signal intensity in mitochondria was very low (data not shown). Further subcellular localization analysis is required by using embryo or endosperm cells.